# SELF-CALIBRATING 4D GAUSSIAN SPLATTING FOR POSE-FREE NOVEL VIEW SYNTHESIS

## ABSTRACT

Dynamic view synthesis (DVS) from monocular videos has remarkably advanced in recent years, achieving high-fidelity rendering with reduced computational costs. Despite these advancements, the optimization of dynamic neural fields still relies on traditional structure from motion (SfM), requiring that all objects remain stationary during scene capture. To address this limitation, we present **SC-4DGS**, a pose-free optimization pipeline for dynamic Gaussian Splatting (GS) from monocular videos, which eliminates the need for SfM through self-calibration. Specifically, we jointly optimize dynamic Gaussian representations and camera poses by utilizing DUSt3R, enabling accurate calibration and rendering. Furthermore, we introduce a comprehensive benchmark, **Kubric-MRig**, that includes extensive camera and object motions along with simultaneous multi-view captures. Unlike previous benchmarks for DVS, where ground truths for camera information are absent due to the difficulty of capturing multiple viewpoints simultaneously, it facilitates evaluating both calibration and rendering quality in dynamic scenes. Experimental results demonstrate that the proposed method outperforms previous pose-free dynamic neural fields and achieves competitive performance compared to existing pose-free 3D neural fields.

## 1 INTRODUCTION

We live in a dynamic world where objects with intricate geometries and textures undergo complex motions and deformations. In daily life, such scenes with motions and deformations are often captured by monocular videos, which do not directly provide the underlying geometries of the scenes. In recent years, computer graphics researchers have explored effective representations and methods to reconstruct 3D scene structures and motions from native visual data. Especially, recent advances in dynamic view synthesis (DVS) (Pumarola et al., 2020; Liu et al., 2023; Yang et al., 2024b) have demonstrated unprecedented fidelity in capturing motions and synthesizing novel views from multi-view input images. The pioneering work, D-NeRF (Pumarola et al., 2020), extends NeRF to learn deformable volumetric field from a set of monocular views without ground truth geometry. To overcome the limited representation power of NeRF, more recent DVS methods tend to use 3D Gaussian Splatting (3DGS) as an alternative representation of scene geometry.

Despite recent advances, existing DVS methods heavily rely on Structure from Motion (SfM), which is susceptible to deformation and motion of objects; for real-world scenes where ground truth camera information is unavailable, the conventional DVS pipeline typically assumes camera information extracted by COLMAP (Schonberger & Frahm, 2016) as ground truth. However, the bundle adjustment process of COLMAP with pair-wise image correspondences often fails to converge. To avoid dependence on SfM, recent approaches (Wang et al., 2021; Jeong et al., 2021; Lin et al., 2021) attempt to jointly optimize camera poses and scene representations, showing successful calibration and rendering quality even when trained without ground truth or COLMAP-extracted camera information. However, they require that all objects remain stationary while capturing videos, which greatly limits their usage in practical scenarios.

To tackle these limitations, we introduce SC-4DGS, an optimization pipeline for pose-free dynamic neural fields. Recent work of RoDynRF (Liu et al., 2023) also jointly estimates camera parameters and neural fields from monocular video in a similar spirit with ours, but optimizing RoDynRF in scenes with extensive camera and object movements is challenging, as the randomly initialized

camera parameters tend to fall into local minima, leading to degraded rendering. To overcome this limitation, SC-4DGS leverages geometric priors from DUSt3R (Wang et al., 2024c), a geometric foundation model for multi-view stereo. To fully take advantage of using DUSt3R, we propose an efficient algorithm for initializing camera poses and the 3D point cloud of 3DGS. Specifically, we introduce batchwise optimization and an extended motion representation tailored for DUSt3R initialization. Additionally, we incorporate physical regularization terms to enable geometrically accurate rendering, which was previously infeasible in RoDynRF due to its fully implicit design.

Furthermore, existing benchmarks encounter difficulties in assessing both calibration and rendering quality because they either lack ground truth (GT) camera poses or simultaneous multi-view captures. Therefore, we introduce a much more challenging benchmark, **Kubric-MRig**, which includes photorealistic scenes with a variety of simultaneously captured viewpoints with extensive camera and object movements. Our experiments show that SC-4DGS outperforms prior pose-free 4D neural fields on Kubric and results competitive performance compared to pose-free 3D neural fields.

In summary, our contributions are as follows:

1. We introduce a pose-free optimization pipeline for dynamic Gaussian Splatting from monocular videos, eliminating the need for Structure from Motion (SfM) through self-calibration.

2. SC-4DGS effectively utilizes geometric priors from DUSt3R by introducing batchwise optimization and an extended motion representation designed for DUSt3R. Additionally, SC-4DGS incorporates regularization terms to ensure geometrically accurate rendering.

3. We introduce a challenging dataset, Kubric-MRig, to evaluate both camera calibration and novel view synthesis performance on dynamic scenes, which was challenging in previous benchmarks.

4. Our optimization pipeline achieves superiority over the pose-free 4D neural fields and competitive performance over previous pose-free 3D neural fields.

## 2 RELATED WORK

### 2.1 NOVEL VIEW SYNTHESIS ON STATIC SCENES

Novel View Synthesis (NVS) is a task of generating novel viewpoints from a set of observations. Pioneer work in NVS leverages point clouds (Kopanas et al., 2021; Zhang et al., 2022; Xu et al., 2022), meshes (Riegler & Koltun, 2020; 2021), and planes (Hoiem et al., 2005) for geometrically convincing view synthesis. Recently, NeRF (Mildenhall et al., 2021) has achieved ground-breaking rendering quality by representing volumetric scene functions via MLPs. To accelerate the training and inference of NeRF, subsequent research has focused on baking trained NeRFs (Hedman et al., 2021) or directly optimizing explicit representations (Fridovich-Keil et al., 2022; Sun et al., 2022; Müller et al., 2022).

More recently, 3D Gaussian Splatting (3DGS) (Kerbl et al., 2023) introduces a novel rendering algorithm that rasterizes anisotropic 3D Gaussians into image planes. Its efficient tile-based alpha-blending CUDA implementation offers real-time rendering with no quality degradation, achieving state-of-the-art results on NVS benchmarks. Subsequent work based on 3DGS has proposed methods to improve fidelity (Kheradmand et al., 2024; Ye et al., 2024), enable training with sparse views (Xiong et al., 2023; Zhang et al., 2024), and facilitate editing (Chen et al., 2024; Dou et al., 2024). Despite their advancements, these approaches assume all objects remain stationary when scene captures and that camera information is fully available, restricting their practical applicability.

### 2.2 NOVEL VIEW SYNTHESIS ON DYNAMIC SCENES

Following the success of NVS in stationary scenes, researchers moved on to extend neural fields for capturing both the underlying motions and geometries of scenes from a set of observations. The pioneer work (Pumarola et al., 2020; Park et al., 2021a;b) learns additional time-varying deformation fields to . Several studies (Li et al., 2022; Fridovich-Keil et al., 2023; Cao & Johnson, 2023) instead learn multi-dimensional feature fields to encode scene dynamics without explicit motion modeling.

With the advent of 3DGS, (Luiten et al., 2024; Wu et al., 2024) propose to learn the trajectories of individual Gaussians over time. Subsequent research has introduced more efficient representations, such as factorized motion bases (Kratimenos et al., 2023) and sparse control points (Huang et al., 2024). Another line of work by (Yang et al., 2024b) extends spherical harmonics into a 4D spherindrical harmonics function, integrating both time-dependent and view-dependent components.

As highlighted by Dycheck (Gao et al., 2022b), many existing approaches focus on unrealistic scenarios, such as camera teleportation or ambient-motion scenes, whereas multi-view capture is typically done using casually captured videos that involve substantial motion. Reconstructing 4D scenes from these videos is a highly ill-posed problem, often failing without additional cues due to the ambiguity between camera and object movements. To resolve the motion ambiguity, recent efforts (Liu et al., 2023; Wang et al., 2024a;b; Lee et al., 2023) leverage pretrained depth estimation models (Ranftl et al., 2020; Yang et al., 2024a) or long-term trajectory tracking models (Karaev et al., 2023). In this study, we tailor DUSt3R (Wang et al., 2024c), a geometric foundation model for initial point clouds and camera poses. For 4DGS optimization, we leverage depth estimation (Yang et al., 2024a) and optical flow (Teed & Deng, 2020) pipelines to ensure geometrically accurate rendering.

### 2.3 POSE-FREE NEURAL FIELDS

Traditional novel view synthesis (NVS) pipelines strongly rely on structure from motion (SfM) (Schonberger & Frahm, 2016) to obtain camera information from a set of observations. Because SfM pipelines are time-consuming and error-prone, researchers are attempting to obtain accurate camera poses without relying on them. There has been growing interests in optimizing neural fields without pre-calibrated camera poses. The pioneer work iNeRF (Yen-Chen et al., 2021) solves an inverse problem that estimates camera poses from pre-trained NeRF by minimizing photometric loss between query views and rendered views. NeRFmm (Wang et al., 2021) and SC-NeRF (Jeong et al., 2021) use photometric loss and geometric regularization to eliminate the required preprocessing step of camera estimation by jointly optimizing camera and NeRF parameters. BARF (Lin et al., 2021) and GARF (Chng et al., 2022) address the gradient inconsistency issue caused by high-frequency parts of positional embeddings to handle complex camera motions. Nope-NeRF (Bian et al., 2023) leverages geometric priors and continuity of camera motions, achieving both high-fidelity rendering and accurate camera trajectory estimation. After the emergence of 3DGS, CF-3DGS (Fu et al., 2024) proposes progressively growing 3DGS for pose estimation. InstantSplat (Fan et al., 2024) shares similar inspiration with our work, leveraging DUSt3R for pose initialization. However, it is designed for static scenes and is restricted to a limited number of viewpoints due to the high memory demands of camera alignment.

RoDynRF, our competitive method, introduces a pose-free optimization pipeline for dynamic scenes by decoupling static backgrounds from dynamic objects. However, it is limited to specific scenarios such as forward-facing scenes or videos with ambient motion. Moreover, its fully implicit representation makes enforcing physical constraints difficult. To address these limitations, our approach employs the geometric foundation model DUSt3R to handle a variety of video capture scenarios. Furthermore, by leveraging the explicit nature of 3DGS, our optimization incorporates geometric regularization to enhance rendering quality.

## 3 METHODS

We present an optimization pipeline that recovers accurate camera poses and time-varying scene geometry from casually captured monocular videos. Specifically, our pipeline processes video frames $I_t \in \mathbb{R}^{H \times W \times 3}$ spanning a total of $F$ frames, jointly optimizing camera poses and dynamic scene representations. Section 3.1 begins with a brief review of the concept of 3D Gaussian Splatting (3DGS)(Kerbl et al., 2023) and the motion representation presented by DynMF(Kratimenos et al., 2023). We then elaborate model details of our SC-4DGS that fully takes advantage of DUSt3R (Wang et al., 2024c) in Section 3.2. Lastly, we introduce several regularization losses to enhance rendering and calibration quality in Section 3.3. The overall optimization pipeline is illustrated in Figure 1.

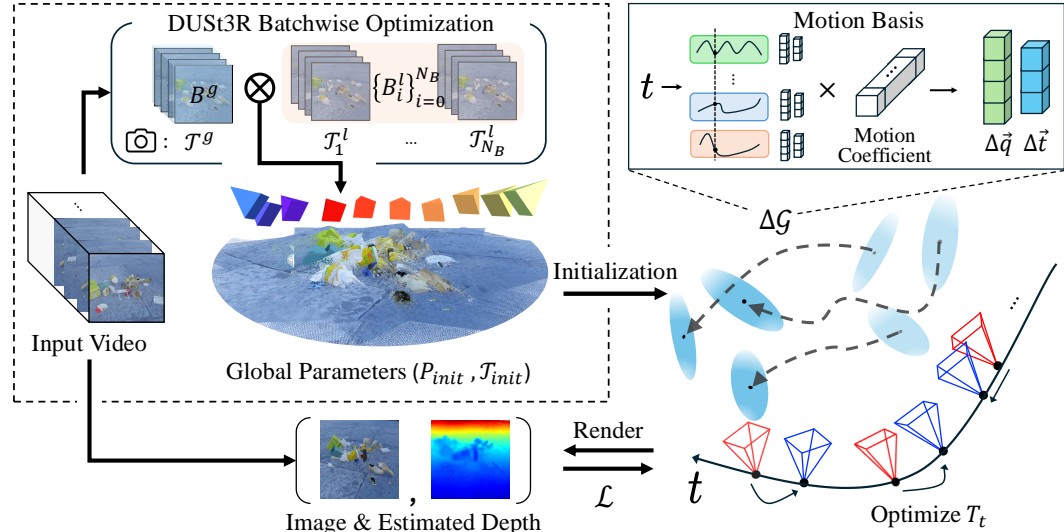

Figure 1: **Overall Pipeline of SC-4DGS.** Given a monocular video input, we estimate the initial camera pose set $\mathcal{T}_{\text{init}}$ and generate an initial point cloud $P_{\text{init}}$. After Dust3R-based optimization, we jointly optimize the dynamic scene representation and the camera poses. The time-dependent transformation of each Gaussian is obtained by combining the outputs of a learnable MLP motion basis and the Gaussian motion coefficients. Parameters mainly optimized using the photometric loss $\mathcal{L}_{recon}$ and the depth loss $\mathcal{L}_{depth}$.

## 3.1 PRELIMINARY: 3D GAUSSIAN SPLATTING AND DYNMF

3D Gaussian Splatting (3DGS) (Kerbl et al., 2023) represents scene geometries using Gaussian primitives and achieves real-time, high-fidelity rendering through an efficient tile-based rasterization. Specifically, each 3D Gaussian is defined by a mean vector $\boldsymbol{\mu}_c$ and a 3D covariance matrix $\boldsymbol{\Sigma}_c$. The influence function at a spatial point $\boldsymbol{x} \in \mathbb{R}^3$ is given by:

$$p(\boldsymbol{x}|\boldsymbol{\mu}_c, \boldsymbol{\Sigma}_c) = e^{-\frac{1}{2}(\boldsymbol{x}-\boldsymbol{\mu}_c)^T \boldsymbol{\Sigma}_c^{-1}(\boldsymbol{x}-\boldsymbol{\mu}_c)}. \tag{1}$$

Then the Gaussians are splatted onto the image plane by approximating (Zwicker et al., 2002) their 2D means and covariances as follows:

$$\boldsymbol{\mu}_c^{2D} = \boldsymbol{\Pi}(KE\boldsymbol{\mu}_c), \quad \boldsymbol{\Sigma}_c^{2D} = \boldsymbol{J}\boldsymbol{E}\boldsymbol{\Sigma}_c\boldsymbol{E}^T\boldsymbol{J}^T, \tag{2}$$

where $\boldsymbol{J}$ denotes the Jacobian of the affine approximation of the projective transformation, and $\boldsymbol{K}$ and $\boldsymbol{E}$ denote intrinsic and extrinsic matrix of camera, respectively. $\boldsymbol{\Pi}$ denotes perspective projection of 3D points into an image plane. Each covariance matrix is decomposed into a rotation matrix $\boldsymbol{R}_c$ and a scaling matrix $\boldsymbol{S}_c$ such that $\boldsymbol{\Sigma}_c = \boldsymbol{R}_c\boldsymbol{S}_c\boldsymbol{S}_c^T\boldsymbol{R}_c^T$, ensuring its semi-positive definiteness. Thus, each Gaussian $\mathcal{G}$ is characterized by mean $\boldsymbol{\mu}$, rotation $\boldsymbol{R}$ and scaling factor $\boldsymbol{S}$, which can be represented as unit quaternion $\boldsymbol{q} \in \mathbb{R}^4$, scaling parameters $\boldsymbol{s} \in \mathbb{R}^3$. 3D Gaussian also includes opacity $\alpha \in \mathbb{R}$ and spherical harmonics(SH) coefficients $\boldsymbol{c} \in \mathbb{R}^{(L+1)^2}$ to represent view-dependent color. The final color of a pixel $\boldsymbol{x}_p$ is computed as:

$$C_p = \sum_{i=1}^{N} c_i \alpha_i p\left(\boldsymbol{x}_p|\boldsymbol{\mu}_c^{2D}, \boldsymbol{\Sigma}_c^{2D}\right) \prod_{j=1}^{i-1}\left(1 - \alpha_j p\left(\boldsymbol{x}_p|\boldsymbol{\mu}_c^{2D}, \boldsymbol{\Sigma}_c^{2D}\right)\right), \tag{3}$$

where $c_i$ and $\alpha_i$ represent the color and opacity associated with each 3D Gaussian.

DynMF (Kratimenos et al., 2023) extends 3DGS to handle dynamic scenes by modeling the trajectory of each Gaussian through learnable motion bases. DynMF defines $B$ shared motion bases predict translation($\boldsymbol{w}^\mu$) and rotation($\boldsymbol{w}^q$) as unit quaternion vector. Each Gaussian has motion coefficients $\boldsymbol{m}$ with a dimension of $B$, time-varying pose of Gaussian is represented by combination of these motion coefficients and the shared motion bases. With a motion bases function $\phi$, DynMF

predicts time-dependent motion of each Gaussian for timestep $t$ as follows:

$$(\boldsymbol{b}^{\mu}(t), \boldsymbol{b}^{q}(t)) = \phi(\frac{t}{T}), \tag{4}$$

$$\boldsymbol{\mu}(t) = \boldsymbol{\mu}_c + \boldsymbol{m} \cdot \boldsymbol{b}^{\mu}(t), \quad \boldsymbol{q}(t) = \boldsymbol{q}_c + \boldsymbol{m} \cdot \boldsymbol{b}^{q}(t). \tag{5}$$

Where $\phi$ is shallow MLP network that receives the normalized timestep in the range $[0, 1]$. Then the time-dependent covariance matrix $\boldsymbol{\Sigma}(t)$ is computed as:

$$\boldsymbol{R}(t) = \text{Q2R}(\boldsymbol{q}(t)), \quad \boldsymbol{\Sigma}(t) = \boldsymbol{R}(t)\boldsymbol{S}\boldsymbol{S}^T \boldsymbol{R}(t)^T, \tag{6}$$

where Q2R denotes a conversion function from quaternions to rotation matrices. Applying the same splatting pipeline with 3DGS, DynMF approximates time-dependent 2D mean $\boldsymbol{\mu}^{2D}(t)$ and covariance $\boldsymbol{\Sigma}^{2D}(t)$ as follows:

$$\boldsymbol{\mu}^{2D}(t) = \boldsymbol{\Pi}(KE\boldsymbol{\mu}(t)), \quad \boldsymbol{\Sigma}^{2D}(t) = \boldsymbol{J}\boldsymbol{E}\boldsymbol{\Sigma}(t)\boldsymbol{E}^T \boldsymbol{J}^T, \tag{7}$$

Finally, the color of pixel $x_p$ at time $t$ is computed as:

$$C_p(t) = \sum_{i=1}^{N} c_i \alpha_i p\left(\boldsymbol{x}_p|\boldsymbol{\mu}(t)^{2D}, \boldsymbol{\Sigma}^{2D}(t)\right) \prod_{j=1}^{i-1} \left(1 - \alpha_j p\left(\boldsymbol{x}_p|\boldsymbol{\mu}^{2D}(t), \boldsymbol{\Sigma}^{2D}(t)\right)\right). \tag{8}$$

In summary, DynMF optimizes three additional components beyond 3DGS: (1) learnable motion bases for quaternion and mean vectors, $\{\boldsymbol{w}_i^q\}_{i=1}^{B}$ and $\{\boldsymbol{w}_i^{\mu}\}_{i=1}^{B}$, (2) a motion coefficient $\boldsymbol{m}$ assigned to each Gaussian, and (3) an MLP network that takes the time $t$ and the motion bases $\boldsymbol{w}^{\mu}$ or $\boldsymbol{w}^q$ as input.

## 3.2 Leveraging DUSt3R for Geometric Priors

In the absence of inherent camera pose priors in monocular videos, previous work has relied on COLMAP (Schönberger et al., 2016) to generate them, though COLMAP is time-consuming and often fails to converge in dynamic scenes. Recently, DUSt3R (Wang et al., 2024c) has shown remarkable performance in real-world settings by training on large-scale 2D-to-3D data. It produces dense per-pixel point maps from two-view inputs with high accuracy, even in dynamic environments. In addition, it supports global alignment through a graph-based optimization for multi-view scenarios. We utilize DUSt3R to initialize the pose and point cloud for 3D Gaussian Splatting. However, DUSt3R's multi-view optimization requires high memory capacity and computational cost, making it unsuitable for temporally densely captured data. Its fully connected graph-based optimization has $O(N^2)$ memory and time complexity because of its pairwise inference, which makes aligning a large number of frames significantly more time-consuming. To overcome this, we introduce an efficient batch-wise optimization pipeline for DUSt3R that effectively acquires camera poses in dense view situations.

**Batchwise Optimization for Efficient Global Alignment** Given a set of frames $\mathcal{F} = \{I_t \in \mathbb{R}^{H \times W \times 3}\}_{t=1}^{F}$, we define two types of optimization batches: a Global Pose Batch $B^g$ and a set of Local Pose Batches $\{B_i^l\}_{i=1}^{N_B}$. Local Pose Batches $B_i^l$ are partitions of frames sequentially sampled from $\mathcal{F}$, with each batch containing $M$ frames, where $N_B = \lceil \frac{F}{M} \rceil$ and $M$ is the sampling stride. We apply the original multi-view alignment of DUSt3R to each local batch independently to obtain the local pose set, $\mathcal{T}_i^l = \{\tilde{\boldsymbol{T}}_k \in SE(3) \mid k = 1, 2, ..., N_B\}$. However, the results from these batches are not aligned within a common global space. To align the results of local optimization, we define $B^g$, which consists of the first images from each local batch and is used to establish the transformations between camera poses in different local pose batches, aligning them in a global space. Using the global pose set $\mathcal{T}^g = \{\boldsymbol{T}_k^g \in SE(3) \mid k = 1, 2, ..., N_B\}$, we can obtain the global camera pose $\boldsymbol{T}_i \in \mathcal{T}$,

$$\boldsymbol{T}_i = \boldsymbol{T}_{\lfloor i/N_B \rfloor}^g \cdot \tilde{\boldsymbol{T}}_{(i \bmod N_B)}, \quad \tilde{\boldsymbol{T}}_{(i \bmod N_B)} \in \mathcal{T}_{\lfloor i/N_B \rfloor}^l. \tag{9}$$

This batchwise strategy reduces the complexity of global alignment to $O(N + N_B)$, while still ensuring efficient alignment across all frames.

**Initializing point cloud with DUSt3R**   After globally aligning all cameras, we generate a point cloud from the DUSt3R point maps, which serves as the initial point cloud $P_{init}$ for training our SC-4DGS. First, we plot all the point clouds obtained from each view's point map in 3D space. Next, we transform each point cloud using the corresponding transformation $\boldsymbol{T}_i \in \mathcal{T}$. Finally, we merge all transformed point clouds into a single global point cloud. After merging, We randomly sample the points with a factor of 0.01.

**Canonicalization of points**   Since our DUSt3R-initialized point cloud comes from various timesteps, we first need to canonicalize all points to the reference timestep $t = 0$. To achieve this, we begin by assigning each Gaussian the timestep from which it originated. Then, for each Gaussian with the assigned timestep $t_i$ the motion value for the target timestep $t$ is adjusted as follows:

$$(\boldsymbol{b}^\mu(t_i, t), \boldsymbol{b}^q(t_i, t)) = \phi(\frac{t}{T}) - \phi(\frac{t_i}{T}).$$

(10)

The rest of the splatting process follows the same steps outlined in Equations 5–8.

### 3.3 OPTIMIZATION

While batch-wise optimization allows us to obtain globally aligned camera poses, slight misalignments still occur. These misalignments arise not only from the inherent inaccuracies of multi-view optimization but also from the inability to utilize information from all images during local batch optimization, leading to minor discrepancies between camera poses from different batches. Our SC-4DGS jointly optimizes neural fields, motion components, and camera poses to further refine the camera poses. To achieve this, we introduce several regularization terms to enforce our model to render high-fidelity images with more accurate camera poses and motions.

**Loss function**   We introduce additional regularization losses beyond those used in 3DGS. Note that 3DGS uses $l1$ reconstruction loss and SSIM loss between rendered and target images:

$$\mathcal{L}_{\text{recon}} = \lambda_{l1}(\|\hat{I}_t - I_t\|_1) + \lambda_{\text{SSIM}}(\frac{1 - \text{SSIM}(\hat{I}_t, I_t)}{2}).$$

(11)

Similar to previous work (Deng et al., 2022; Turkulainen et al., 2024), we employ a photometric reconstruction loss along with geometric priors to address ambiguities arising from limited observations when reconstructing time-varying geometry.

First, we regularize the underlying geometries of scenes using monocular depth maps obtained from DepthAnything (Yang et al., 2024a). However, due to the scale ambiguity of the predicted monocular depths, we cannot directly compare the estimated depth with the rendered scene depth. To address this issue, we apply the Pearson depth loss $\mathcal{L}_{\text{depth}}$ (Xiong et al., 2023), which maximizes the linear correlation between the rendered depth and the estimated depth. $\mathcal{L}_{\text{depth}}$ is designed to maximize the PCC between the rendered depth map $\hat{D}_t$ and the estimated depth $D_t$ by DepthAnything as follows:

$$\mathcal{L}_{\text{depth}} = \frac{1}{N} \sum_{t=1}^{N_F} \left(1 - \mathcal{E}(\hat{D}_t, D_t)\right), \quad \mathcal{E}(\hat{D}_t, D_t) = \frac{\mathbb{E}[\hat{D}_t D_t] - \mathbb{E}[\hat{D}_t]\mathbb{E}[D_t]}{\sigma[\hat{D}_t] \cdot \sigma[D_t]},$$

(12)

where $\sigma$ is the standard deviation function. Note that the Pearson correlation coefficient(PCC), $\mathcal{E}(\hat{D}_t, D_t)$, measures the cross-correlation between $X$ and $Y$. We compute two types of depth loss-global depth loss $\mathcal{L}_{depth,g}$ and local depth loss $\mathcal{L}_{depth,l}$ to compare local statistics, which remove local noise of depth.

Second, following the regularization from DynMF (Kratimenos et al., 2023), we apply the motion coefficient sparsity regularization losses $\mathcal{L}_m$ and $\mathcal{L}_{ms}$. These losses encourage the motion coefficients to be sparse, which helps prevent overfitting to perturbations and noisy motions of training viewpoints. Formally, they are defined as:

$$\mathcal{L}_m = \frac{1}{NB} \sum_{i=1}^{N} \sum_{j=1}^{B} \|m_{ij}\|, \quad \mathcal{L}_{ms} = \frac{1}{N} \sum_{i=1}^{N} \left(\frac{1}{B} \sum_{j=1}^{B} \frac{|m_{ij}|}{\max_{1 \le k \le B} |m_{ik}|}\right).$$

(13)

In addition, since the motion bases receive only temporal information and do not account for the spatial locality of each Gaussian's motion, Gaussians that are spatially close often represent the same rigidly moving object, leading to strongly correlated motions over time. To enforce this spatial coherence, we introduce a rigidity loss applied to the motion coefficients $\boldsymbol{m}_i$ and $\boldsymbol{m}_j$ of the $i$-th and $j$-th Gaussian, defined as:

$$\mathcal{L}_{\text{rigid}} = \frac{1}{Nk} \sum_{i=1}^{N} \sum_{j \in \text{NN}(\mathcal{G}_i)} \exp\left(-\lambda_w \|\boldsymbol{\mu}_i - \boldsymbol{\mu}_j\|_2^2\right) \|\boldsymbol{m}_i - \boldsymbol{m}_j\|^2. \tag{14}$$

This loss is applied to the $k$ nearest neighbors of the $i$-th Gaussian, $\mathcal{G}_i$. Thus, the total motion loss is defined as follows:

$$\mathcal{L}_{\text{motion}} = \lambda_{\text{rigid}}\mathcal{L}_{\text{rigid}} + \lambda_m \mathcal{L}_m + \lambda_{ms}\mathcal{L}_{ms}, \tag{15}$$

where $\lambda_{\text{rigidity}}$, $\lambda_1$, and $\lambda_s$ are hyperparameters controlling the influence of each loss term.

Third, we encourage the static parts of scenes to have the same rendered results across different timesteps. To achieve this, we apply a reconstruction loss for randomly sampled timestep $\hat{t}$ from $[0, 1)$ to the rendered image $\hat{I}_{\hat{t}}$ and the target image $I_t$ using a static mask $M$, which is obtained based on epipolar errors.

$$\mathcal{L}_{\text{static}} = \|\hat{I}_t[M] - I_t[M]\|_1, \text{where } t \sim [0, 1). \tag{16}$$

The detailed process of obtaining the static mask is provided in the Appendix A.2.2.

Lastly, we enforce the smoothness of camera trajectories, since this characteristic is typically observed in videos captured by handheld devices. Intuitively, we can use the constant speed assumption commonly applied in many SLAM pipelines. For camera pose $\boldsymbol{T}_t$ in timestep $t$, we can apply first-order motion regularization loss $\mathcal{L}_{\nabla,t}$:

$$\mathcal{L}_{\nabla,t} = \|\Delta\boldsymbol{T}_t - \Delta\boldsymbol{T}_{t-1}\|_1 = \|\boldsymbol{T}_t - 2\boldsymbol{T}_{t-1} + \boldsymbol{T}_{t-2}\|_1, \tag{17}$$

where pose $\boldsymbol{T}_t = (\boldsymbol{t}_t, \boldsymbol{q}_t)$ includes translation and querternion vector. And $\Delta\boldsymbol{T}_t = \boldsymbol{T}_t - \boldsymbol{T}_{t-1}$ is first-order difference of camera pose at time $t$. However, this first-order pose regularization can overly constrain the camera trajectory to a linear form. To relax this condition, we apply second-order pose regularization loss $\mathcal{L}_{\nabla^2,t}$ to the camera trajectory:

$$\mathcal{L}_{\nabla^2,t} = \|\Delta^2\boldsymbol{T}_t - \Delta^2\boldsymbol{T}_{t-1}\|_1 = \|\boldsymbol{T}_t - 3\boldsymbol{T}_{t-1} + 3\boldsymbol{T}_{t-2} - \boldsymbol{T}_{t-3}\|_1. \tag{18}$$

We apply this pose regularization loss for all frames as follows:

$$\mathcal{L}_{\text{cam}} = \sum_t \mathcal{L}_{\nabla^2,t}. \tag{19}$$

This loss encourages smooth transitions in camera motion, preventing sudden changes while providing more flexibility than first-order regularization.

Thus, our final loss is the joint loss the introduced losses:

$$\mathcal{L} = \mathcal{L}_{\text{recon}} + \lambda_{\text{depth,g}}\mathcal{L}_{\text{depth,g}} + \lambda_{\text{depth,l}}\mathcal{L}_{\text{depth,l}} + \lambda_{\text{motion}}\mathcal{L}_{\text{motion}} + \lambda_{\text{static}}\mathcal{L}_{\text{static}} + \lambda_{\text{cam}}\mathcal{L}_{\text{cam}}. \tag{20}$$

## 4 EXPERIMENTS

### 4.1 DATASET: KUBRIC-MRIG

We revisit previous benchmarks–Tanks and Temples (Knapitsch et al., 2017), D-NeRF (Pumarola et al., 2020), NVIDIA dynamic (Gao et al., 2022a), Nerfies-HyperNeRF (Park et al., 2021b), and iPhone (Yoon et al., 2020)– on novel view synthesis(NVS) to assess the suitability for estimating calibration and NVS performance for dynamic scenes. As summarized in Table 1, the previous benchmarks have several limitations: they offer restricted viewpoints such as forward-facing scenes (Gao et al., 2022a; Yoon et al., 2020), feature no or only ambient motion (Knapitsch et al., 2017; Gao et al., 2022a), or lack ground truth camera poses (Knapitsch et al., 2017; Gao et al., 2022a; Yoon et al., 2020). To address these limitations, we introduce Kubric-Mrig, a dataset specifically designed

| Dataset | Wide Viewpoints | Large Motion | GT CAM | Backgrounds |
|---------|:---------------:|:------------:|:------:|:-----------:|
| T & T | ✓ | ✗ | ✗ | ✓ |
| D-NeRF | ✓ | ✓ | ✓ | ✗ |
| iPhone | ✗ | ✗ | ✗ | ✓ |
| Nerfies-HyperNeRF | ✗ | ✓ | ✗ | ✓ |
| NVIDIA | ✗ | ✓ | ✓ | ✓ |
| Kubric-MRig (ours) | ✓ | ✓ | ✓ | ✓ |

Table 1: **Summary of previous benchmarks for pose-free dynamic novel view synthesis.** Previous benchmarks either lack wide viewpoints, large motions, ground truth cameras, or complex backgrounds.

to evaluate both calibration and NVS performance for dynamic scenes with large movements of cameras and objects.

In detail, we use the Kubric (Greff et al., 2022) engine, a Blender-based synthetic scene generator, to create the Kubric-MRig dataset. For training, we generate monocular videos by moving the cameras around the objects and capturing viewpoints over 100 incremental timesteps. For evaluation, we introduce two types of evaluation setups: pose-freeze view-change and view-freeze time-varying. In the pose-freeze view-change setup, the camera position is fixed at the first view from the training set, and the timestep varies across the 100 timesteps used for training. In contrast, the view-freeze time-varying setup keeps the timestep fixed at 0, while the viewpoints are set to those used during training. For more detailed information, please refer to the Appendix A.1.

## 4.2 POSE-FREE DYNAMIC NOVEL VIEW SYNTHESIS

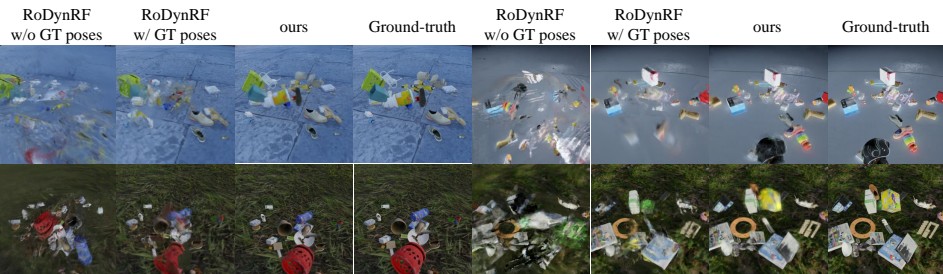

Figure 2: **Qualitative results on Kubric-MRig.** Our pipeline accurately reconstructs scene geometry, produces sharp renderings, and aligns object positions well. Without GT camera poses, RoDynRF struggles to learn the scene geometry, resulting in object positions that differ from the GT. Even with GT camera poses, RoDynRF produces blurry results.

| | GT CAM | PSNR(↑) | SSIM(↑) | LPIPS(↓) | ATE(↓) | RPE-R(↓) | RPE-t(↓) |
|---|:------:|:-------:|:-------:|:--------:|:------:|:--------:|:--------:|
| D-NeRF | ✓ | 19.65 | 0.6692 | 0.4377 | - | - | - |
| RoDynRF | ✓ | 20.27 | 0.7514 | 0.4838 | - | - | - |
| 4DGS1 | ✓ | 20.78 | 0.7005 | 0.3984 | - | - | - |
| 4DGS2 | ✓ | 21.65 | 0.8415 | 0.1974 | - | - | - |
| Deform3D | ✓ | 21.73 | 0.8365 | 0.2146 | - | - | - |
| RoDynRF | ✗ | 18.10 | 0.6180 | 0.6038 | 0.0632 | 0.4088 | 1.8255 |
| ours | ✗ | **19.19** | **0.6346** | **0.4615** | **0.0039** | **0.2399** | **0.0608** |

Table 2: **Comparison of NVS and calibration performance on Kubric-MRig with dynamic neural fields.** GT CAM denotes the availability of ground truth camera information when training models. Our SC-4DGS achieves superiority over RoDynRF for both rendering and calibration quality.

We compare our SC-4DGS with previous dynamic neural fields on Kubric-MRig. Following the evaluation protocol of (Fu et al., 2024), we assess visual quality using PSNR, SSIM, and LPIPS, and calibration quality using ATE, RPE-R, and RPE-t, with detailed explanations of each metric and implementation details of our pipeline provided in the Appendix. As shown in Table2, our SC-4DGS outperforms the previous pose-free dynamic neural field, RoDynRF (Liu et al., 2023), in both NVS and calibration performance. Specifically, SC-4DGS shows a significant improvement in calibration quality over RoDynRF.

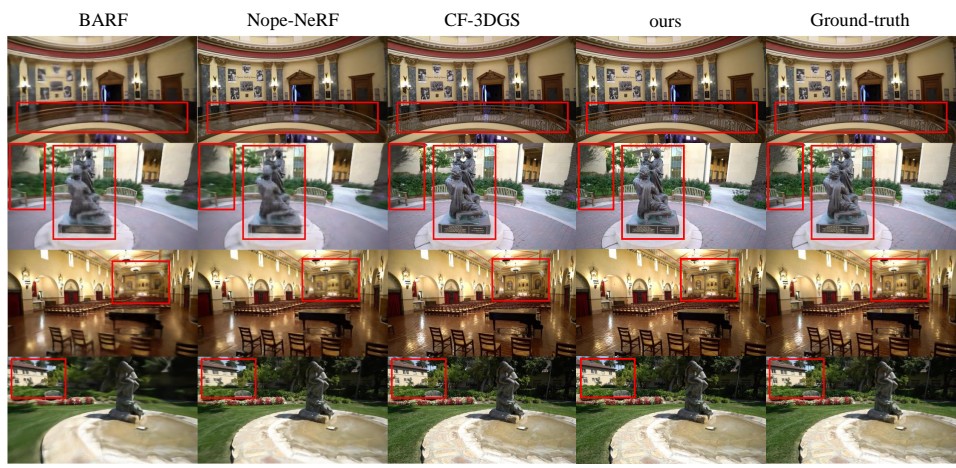

Figure 3: **Qualitative results on Tanks and Temples.** We show rendering results that are more realistic than other baselines, and comparable to CF-3DGS.

| | PSNR(↑) | SSIM(↑) | LPIPS(↓) | ATE(↓) | RPE-R(↓) | RPE-t(↓) |
|---|---|---|---|---|---|---|
| NeRFmm | 22.50 | 0.59 | 0.54 | 0.123 | 0.477 | 1.735 |
| SC-NeRF | 23.76 | 0.65 | 0.48 | 0.129 | 0.489 | 1.890 |
| BARF | 23.42 | 0.61 | 0.54 | 0.078 | 0.441 | 1.046 |
| Nope-NeRF | 26.34 | 0.74 | 0.39 | 0.006 | 0.038 | 0.080 |
| CF-3DGS | **31.28** | **0.93** | **0.09** | **0.004** | 0.069 | **0.041** |
| ours | 31.07 | 0.91 | 0.10 | 0.006 | **0.028** | 0.053 |

Table 3: **Comparison of pose-free NVS methods.** Quantitative results of calibration performance on Tanks and Temples with static pose-free neural fields. Ours achieves competitive performance with CF-3DGS while showing notable superiority over other methods.

We also evaluate other dynamic neural fields—D-NeRF (Pumarola et al., 2020), RoDynRF (Liu et al., 2023), 4DGS1 (Yang et al., 2024b), 4DGS2 (Wu et al., 2024), and Deform3D (Yang et al., 2024c)—when ground truth (GT) camera poses are available. While SC-4DGS still requires further improvements to match the performance of methods with access to GT poses, it is important to note that GT poses are often unavailable in practical scenarios due to the limitations of structure-from-motion (SfM) methods in handling object motions and deformations. As shown in Figure 2, RoDynRF struggles to render accurately, whereas SC-4DGS produces much clearer renderings.

### 4.3 POSE-FREE STATIC NOVEL VIEW SYNTHESIS

Due to the limited baselines in pose-free dynamic view synthesis, we also compare our model with previous pose-free static neural fields—NeRFmm (Wang et al., 2021), SC-NeRF (Jeong et al., 2021), BARF (Lin et al., 2021), Nope-NeRF (Bian et al., 2023), and CF-3DGS (Fu et al., 2024)—on the Tanks and Temples dataset (Knapitsch et al., 2017). For a fair comparison, we disable the motion learning components to adapt to static scenes. We follow the same evaluation pipeline with CF-3DGS to align test poses. As shown in Table 3, SC-4DGS demonstrates comparable rendering and calibration quality to the previous state-of-the-art pose-free static neural field, CF-3DGS, while achieving a notable improvement in RPE-R over CF-3DGS.

### 4.4 ABLATION STUDY

We conduct control experiments to evaluate the impact of each component of our work.

**Pose Initialization Strategies** We examine various pose initialization methods on Kubric-MRig. We exclude COLMAP from the comparison, as it frequently fails to converge in dynamic scenes. We explore four batch sampling strategies when initializing poses via DUSt3R: naive, sequential (SQ), strided batch (SB), and our proposed method. The naive strategy simply accumulates all pair-wise predictions, the SQ strategy create local batch ands connect the last of previous batch and the

first one, and the SB uses strided batch technique for optimization. For more details, please refer to Appendix 4.4.

As shown in Table 4, the naive strategy produces pair-wise predictions with inconsistent scale across multiple views, resulting in significant pose errors. The SB strategy performs better than the naive approach but is still vulnerable to object motion due to the large timestep between frames in each batch. According to Table 5, while the SQ strategy achieves better RPE scores than our approach, it results in worse visual quality when used for pose initialization. We have selected our current strategy as it offers a better balance between NVS performance and pose estimation quality.

| | ATE($\downarrow$) | RPE-R($\downarrow$) | RPE-t($\downarrow$) |
|---|---|---|---|
| DUSt3R (naive) | 0.0594 | 2.130 | 0.5396 |
| DUSt3R (SQ) | 0.0071 | **0.5429** | **0.1211** |
| DUSt3R (SB) | 0.0044 | 2.533 | 0.4210 |
| ours | **0.0038** | 0.7263 | 0.1616 |

Table 4: **Comparison of DUSt3R optimization strategy.** We report the pose estimation performance for each DUSt3R batchwise optimization strategy.

**Regularization**   We also conduct ablation studies of the regularization terms to evaluate the impact of each component on the scene0 of Kubric-MRig. Specifically, we exclude pose difference, depth, rigidity, and motion regularization from our model. We also examine the effect of replacing the second-order pose regularization with first-order pose regularization, $\mathcal{L}_{cam,\nabla}$. Without pose regularization, it shows higher SSIM score even though the pose quality significantly degrades.

| | PSNR($\uparrow$) | SSIM($\uparrow$) | LPIPS($\downarrow$) | ATE($\downarrow$) | RPE-R($\downarrow$) | RPE-t($\downarrow$) |
|---|---|---|---|---|---|---|
| ours | **21.40** | 0.6449 | 0.4548 | **0.0025** | 0.3508 | 0.0690 |
| use $\mathcal{L}_{cam,\nabla}$ | 20.18 | 0.6410 | 0.5426 | 0.0075 | 0.4164 | 0.1156 |
| SQ pose init. | 20.77 | 0.5548 | 0.4763 | 0.0037 | **0.2601** | **0.0513** |
| w/o $\mathcal{L}_{cam}$ | 21.02 | **0.6587** | 0.5351 | 0.0032 | 1.1379 | 0.2544 |
| w/o $\mathcal{L}_{static}$ | 21.22 | 0.6359 | 0.5047 | **0.0025** | 0.3342 | 0.0669 |
| w/o $\mathcal{L}_{rigid}$ | 21.02 | 0.6349 | 0.4909 | **0.0025** | 0.3381 | 0.0671 |
| w/o $\mathcal{L}_{depth}$ | 21.00 | 0.6318 | **0.4541** | **0.0025** | 0.3386 | 0.0665 |

Table 5: **Ablation studies.** Result of ablation studies on different regularization terms and pose initialization methods, evaluating rendering and calibration quality on Kubric-MRig scene0.

According to Table 5, our method demonstrates the best performance in terms of PSNR and ATE, indicating precise camera calibration and high-quality rendering. When we replace the camera motion regularization with the first-order loss $\mathcal{L}_{cam,\nabla}$, the performance degrades, highlighting the effectiveness of the second-order camera regularization. Removing the camera regularization term $\mathcal{L}_{cam}$ leads to significantly worse pose optimization results. Excluding the static regularization $\mathcal{L}_{static}$ or the rigidity regularization $\mathcal{L}_{rigid}$ causes a noticeable decrease in PSNR and an increase in LPIPS values. These losses are crucial for accurately modeling the dynamic and static components separately, playing important roles in our pipeline. Additionally, excluding the depth regularization $\mathcal{L}_{depth}$ slightly reduces rendering quality, emphasizing the contribution of depth information in enhancing the final results.

## 5   CONCLUSION

In this paper, we introduced SC-4DGS, a camera-free optimization pipeline for dynamic Gaussian Splatting (GS) from monocular videos. Our method addresses the limitations of existing dynamic view synthesis (DVS) models, which still heavily rely on structure from motion (SfM) and assume static scenes during capture. By fully exploiting geometric priors from geometric foundational models, SC-4DGS achieves geometrically accurate and high-quality rendering in dynamic scenes without requiring ground truth camera information. Additionally, we proposed Kubric-MRig, a challenging benchmark designed to evaluate both calibration and novel view synthesis performance under extensive object and camera motions. SC-4DGS demonstrates superior performance over previous pose-free dynamic neural fields and achieves competitive results when compared to existing pose-free 3D neural fields, marking a significant step forward in the optimization of dynamic neural fields.

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

## A    APPENDIX

### A.1    KUBRIC-MRIG DATASET

We provide further details on the process of generating the Kubric-MRig dataset. Our data generation script is based on the Movi script, which is part of the official implementation of Kubric (Greff et al., 2022). We randomly select 10 to 20 static objects and 1 to 3 dynamic objects from the Google Scanned Objects dataset. We then choose a background from the publicly available HDRI environments in Kubric. The static objects are randomly placed on the ground, while the dynamic objects are positioned to float in the air. Next, we run a physics simulation to achieve realistic object movements, resembling real-world scenarios.

For the training set, we deploy 100 cameras that follow circular trajectories around the objects, with equal spacing between each frame to ensure consistent scene coverage. For the evaluation set, we use the same camera positions as in the training setup, but with two distinct evaluation scenarios: "pose-freeze time-varying" and "time-freeze pose-varying". In the "pose-freeze time-varying" setup, we fix the camera viewpoint to the first training camera position, then capture the scene across the 100 timesteps used during training. In the "time-freeze pose-varying" setup, we fix the timestep to 0 and capture the scene from the same viewpoints used in the training data.

All cameras are positioned equidistant from the world center, with distances randomly sampled between 15 and 20 units. To ensure the viewpoint coverage of the scenes for evaluation, we fix the elevation angle, which is randomly sampled from $30°$ to $60°$ during data capture. We provide visualization of the generated dataset in Figure 4.

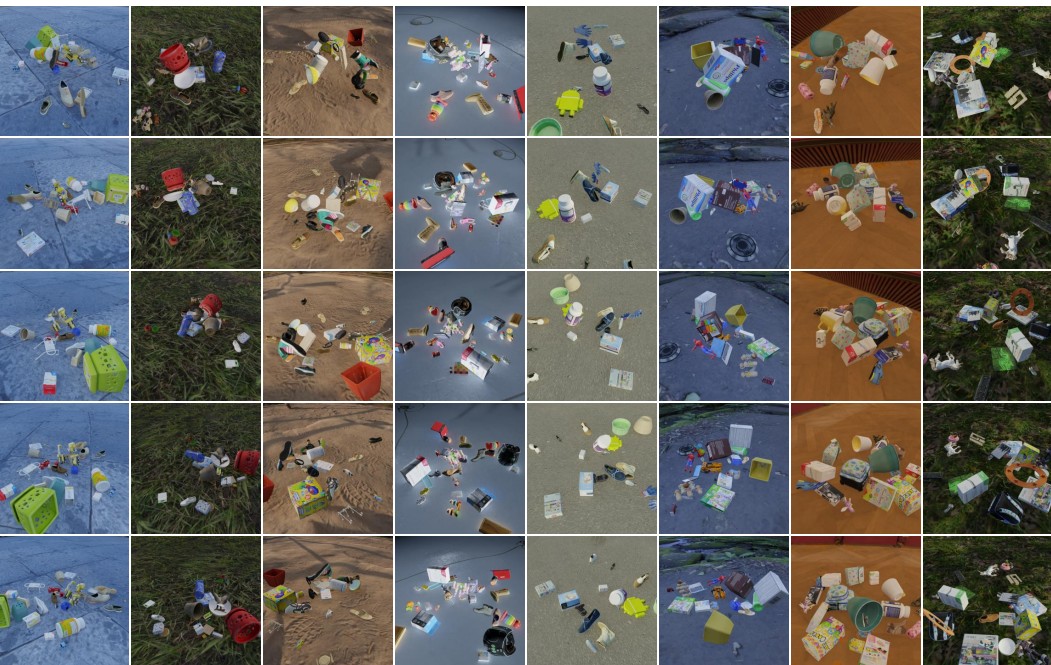

Figure 4: **Visualization of samples from the Kubric-MRig.**

### A.2    IMPLEMENTATION DETAILS

#### A.2.1    OPTIMIZATION

**4DGS Optimization.**    We use the official implementation for 3DGS with adding gradient computation over camera poses. When running DUSt3R batchwise optimization process to get initial camera poses, we use the window size $M$ of 10 and set the number of iteration to 250 on each batch. After optimization, we uniformly sample points from globally aligned point cloud of all viewpoints with a factor of 0.01 to initialize point cloud for GS. To extract monocular depths and optical flows,

we use DepthAnything(Yang et al., 2024a) and RAFT(Teed & Deng, 2020). When optimizing SC-4DGS, we set the loss weights as follows: $\lambda_{\text{SSIM}} = 0.2$, $\lambda_{l1} = 0.8$, $\lambda_m = 0.2$, $\lambda_{ms} = 0.05$, $\lambda_{\text{depth,g}} = 0.15$, $\lambda_{\text{depth,l}} = 0.05$, $\lambda_{\text{static}} = 1.0$, and $\lambda_{\text{cam}} = 0.1$. Additionally, rigidity loss is applied every 5 iterations with $\lambda_{\text{rigid}} = 0.2$. Similar to DynMF, for the first 5000 steps, we do not optimize motion networks for stable training. Then, we linearly warm-up learning rates for the 10% of the total steps and anneal with cosine functions for the rest of iterations. We set the peak learning rate of camera rotation to 0.0001 and translation to 0.0002. For the rest of configurations, we follow the official implementation of GS.

**3DGS Optimization.** In contrast to the Kubric-MRig dataset, for the Tanks and Temples dataset, we have disabled the learning of Gaussian motion. As a result, we obtained camera poses from the DUSt3R batch-wise optimization process using the SQ strategy. Therefore, we optimized the Gaussians and camera poses without applying the pose regularization process and set the loss weights as follows: $\lambda_{\text{SSIM}} = 0.2$, $\lambda_{l1} = 0.8$, $\lambda_{\text{depth,g}} = 0.001$, and $\lambda_{\text{depth,l}} = 0.01$. We also applied cosine annealing throughout the iterations and set the learning rates for camera rotation and translation to 0.000005 and 0.00005, respectively. During training, we optimized the Gaussians and camera poses without resetting opacity, without learning motion, and without applying the pose regularization process.

### A.2.2   Obtaining Motion Masks with Epipolar Errors

RoDynRF (Liu et al., 2023) utilizes RAFT (Teed & Deng, 2020) to first predict forward and backward optical flows from video frames. Then, it estimates the fundamental matrix between adjacent frames using the 8-point algorithm. Afterward, it computes the error between the points projected using the fundamental matrix and those derived from the predicted flows. Regions with high error are assumed to correspond to dynamic parts of the frames. However, in our Kubric-MRig dataset, this method often fails due to the large motions observed between adjacent frames. To mitigate this issue, we directly compute the fundamental matrix using the calibrated poses during training. We then assume that regions with an epipolar error below the median correspond to the static parts of the scene.

### A.2.3   DUSt3R Optimization Strategies

In ablation studies, we have proposed three additional strategies to optimize camera poses of dense views. Here, we elaborate details of each strategy. In the naive strategy, we sequentially accumulate pair-wise predictions without any extra optimization. For the SQ strategy, we sample 10 consecutive frames from the entire sequence to create each local batch, with the final batch containing the remaining frames. After running DUSt3R pose optimization on each local batch, we align the last frame of the $N$-th local batch with the first frame of the $(N+1)$-th batch. Lastly, for the SB strategy, we first sample the first $K$ frames (where $K$ is the quotient of the total number of frames divided by 10) to form a global batch, and then sample every 10th frame starting from each frame in the global batch to form local batches. After running DUSt3R on all batches, we align each local batch with the corresponding frames in the global batch. Remark that our strategy samples every 10th frame starting from the first frame of the entire sequence to form the global batch. For each local batch, we sample frames sequentially, starting from the $K$-th frame to the $(K+1)$−th frame in the global batch. We then run DUSt3R pose optimization on each batch and align each local batch with the corresponding frames in the global batch.

### A.3   Failure Cases and Future Work

**Failure Cases.**   While our model outperforms existing baselines on the Kubric-MRig dataset, it has several limitations where failure cases can occur. One major limitation is that our model cannot handle temporally sparse videos with large camera motions. Such videos provide insufficient observations, leading to potential optimization failures. Additionally, our pose regularization may oversmooth sparse camera viewpoints with significant pose variations, resulting in suboptimal performance in these settings.

Another limitation is that our pipeline is vulnerable to failures in Dust3R optimization. In scenarios like dynamic scenes with textureless backgrounds, Dust3R optimization can fail similarly to other SfM pipelines(Schonberger & Frahm, 2016), causing the subsequent training process to fail entirely.

In some cases, the camera poses optimized by Dust3R lead the training to fall into local minima. Slight misalignments between camera poses across frames result in insufficient gradients, hindering further optimization. These issues indicate that the robustness of our method is contingent on the success of Dust3R optimization, highlighting the need for mechanisms to handle or mitigate such failures to enhance overall reliability.

**Future Work.**    Additionally, to handle in-the-wild videos, we aim to integrate static and dynamic scene training procedures. Our current approach distinguishes between static and dynamic scenes by disabling motion learning when training on static scenes. However, separating dynamic and static settings is unrealistic for handling diverse in-the-wild videos. Real-world scenes often contain both static and dynamic elements, and our model's inability to seamlessly integrate both limits its applicability. This suggests the need for a unified framework that can adaptively handle both static and dynamic components within a scene.

