# OpenReview forum: "Towards Pose-Free Dynamic Neural Fields: Leveraging Geometric Foundation Models"
_ICLR.cc/2025/Conference — ICLR 2025 Conference Withdrawn Submission_

### Official Review · Reviewer_8aze · 2024-10-23

**Soundness:** 2
**Presentation:** 3
**Contribution:** 2
**Rating:** 3
**Confidence:** 5

**Summary:**

This paper is built on the existing work DUSt3R. Firstly, the authors implemented DUSt3R with their new way of batch-wise alignment for the point clouds and camera pose initialization. After the canonicalization of points, with the benefit of monocular depth estimation from DepthAnything and the motion mask estimation from RoDynRF, they introduce more regularization loss to jointly optimize the camera poses, and scene representations. In addition, they claimed to propose a more challenging dataset for DVS. However, I keep my doubts about the statement that the proposed dataset is challenging. Besides, their heavy reliance on the initialization of DUSt3R makes me doubt their contribution.

**Strengths:**

1. The authors incorporate DUSt3R for a wonderful point cloud and camera pose initialization.
2. The authors proposed a new dataset that can benefit the research in this field.
3. The author provides experiments on both dynamic and static scenes.

**Weaknesses:**

1. From my perspective, this paper lacks comparisons with proper baselines. For example, there are some accepted works like casualSAM[1], LEAP-VO[2], ParticlesfM[3], ..., which I think should be compared with. Or can the authors provide some reasons for why lack of them?
2. I think the authors should include the comparisons with COLMAP, since although COLMAP is somehow theoretically designed for static scenes, I believe it does not completely fail on most of the datasets. The exhaustive matching step can help COLMAP get rid of some (dynamic) outliers, even though some noise exists.
2. I do not agree that in table 1, the authors claimed that the iPhone dataset does not contain large motion (e.g. 'spin', 'apple', 'space-out', 'pillow', 'teddy', ...).
3. From my perspective, the authors introduce a lot of regularization terms, however, for dynamic scene experiments, the authors only conducted experiments on one public dataset, which I think is not enough.
4. The authors claimed that COLMAP is very time-consuming, but I do not see any time comparisons with COLMAP or the existing baselines.


[1] Zhang, Zhoutong, Forrester Cole, Zhengqi Li, Michael Rubinstein, Noah Snavely, and William T. Freeman. "Structure and motion from casual videos." In European Conference on Computer Vision, pp. 20-37. Cham: Springer Nature Switzerland, 2022.

[2] Chen, Weirong, Le Chen, Rui Wang, and Marc Pollefeys. "LEAP-VO: Long-term Effective Any Point Tracking for Visual Odometry." In Proceedings of the IEEE/CVF Conference on Computer Vision and Pattern Recognition, pp. 19844-19853. 2024.

[3] Zhao, Wang, Shaohui Liu, Hengkai Guo, Wenping Wang, and Yong-Jin Liu. "Particlesfm: Exploiting dense point trajectories for localizing moving cameras in the wild." In European Conference on Computer Vision, pp. 523-542. Cham: Springer Nature Switzerland, 2022.

**Questions:**

1. Since not mentioned in the paper, did you use the G.T. camera intrinsic? Or optimizing the camera intrinsic? Please make it clear in the paper.
2. The proposed method seems to highly rely on DUSt3R which is an existing work for static scenes. But in the method section, the author claimed they used DUSt3R for camera pose initialization, I agree that DUSt3R can be a good initialization in some small or regular motion scenes, but I am wondering whether DUSt3R can work well in some large movement datasets such as DAVIS[1], iPhone,...? I am expecting the authors can conduct more experiments on DAVIS and iPhone to show the effectiveness and robustness of their method.
3. Can the authors provide some motion mask samples generated by them? Since I think the motion mask estimation of RoDynRF is not that robust and effective.

[1] Pont-Tuset, Jordi, Federico Perazzi, Sergi Caelles, Pablo Arbeláez, Alex Sorkine-Hornung, and Luc Van Gool. "The 2017 davis challenge on video object segmentation." arXiv preprint arXiv:1704.00675 (2017).

---

### Official Review · Reviewer_p8SA · 2024-11-03

**Soundness:** 2
**Presentation:** 4
**Contribution:** 2
**Rating:** 5
**Confidence:** 4

**Summary:**

This paper addresses the limitations of traditional dynamic view synthesis (DVS) from monocular videos, which relies on Structure from Motion (SfM) and thus requires stationary scenes. To overcome this, the authors propose SC-4DGS, a pose-free optimization pipeline for dynamic Gaussian Splatting (3DGS) that eliminates the need for SfM through self-calibration using geometric priors from DUSt3R. SC-4DGS introduces batchwise optimization and an extended motion representation to jointly optimize camera poses and scene representations in dynamic settings. Additionally, the paper introduces Kubric-MRig, a benchmark dataset with photorealistic scenes featuring complex camera and object movements for evaluating calibration and rendering quality. Experiments show that SC-4DGS outperforms existing pose-free 4D neural fields and performs competitively with 3D neural fields, proving effective for complex, real-world dynamic scenes.

**Strengths:**

1. The paper is among the first to achieve pose-free 4D Gaussian Splatting, which provides valuable insights for future research in dynamic scene reconstruction.
2.The paper provides highly detailed mathematical formulations, which effectively build the theoretical foundation of SC-4DGS. Also, this thorough mathematical foundation enhances clarity, helping readers understand both the model structure and the optimization processes involved.
3.The introduction of the Kubric-MRig dataset fills a gap in DVS benchmarking by incorporating extensive camera and object motion alongside ground truth data for calibration and novel view synthesis, providing a robust dataset for dynamic scene evaluation.
4.The SC-4DGS method demonstrates large improvements over previous dynamic neural field approach.

**Weaknesses:**

1.The SC-4DGS model lacks originality, as it largely combines the existing DYNMF and DUSt3R models. The primary change appears to be the replacement of COLMAP with DUSt3R for point cloud generation, without significant structural improvements. Demonstrating unique contributions or optimizations beyond this combination would strengthen the paper’s impact.
2. Although the paper highlights the high computational complexity of DUSt3R, it lacks specific training time comparisons. Quantitative results showing the time reduction achieved through the paper’s optimizations would help illustrate the efficiency of SC-4DGS.
3.The experiments mainly compare SC-4DGS with RoDyNeRF under conditions without ground-truth (GT) camera poses, and the results do not surpass methods using GT. This limited comparison, with only RoDyNeRF as a baseline in the absence of GT, raises concerns about the reliability of the findings.
4.The paper states that most methods use COLMAP to serve as GT camera poses and point cloud; however, COLMAP estimates are not true GT values but approximations. Since SC-4DGS replaces COLMAP with DUSt3R for point cloud and pose estimation, it’s essential to demonstrate the advantages of this change. An experiment comparing COLMAP’s and DUSt3R’s effectiveness in scenarios where COLMAP fails but DUSt3R succeeds would substantiate SC-4DGS’s advantage.
5.Although SC-4DGS is designed for monocular video inputs, the Kubric-MRig dataset features wide variation among its 100 cameras, and rapid camera switching may lead the monocular sequence to resemble stereo video. This could create inconsistencies when using the dataset to evaluate the performance of a strictly monocular input model. Adjustments to the dataset or a careful clarification of how it aligns with the paper’s objectives would address these concerns.

**Questions:**

1.Based on my understanding, RoDyNeRF uses an explicit neural voxel radiance field in the section 3. The paper’s description of it as an implicit representation is inaccurate and could lead to misunderstandings.
2.In Table 1, the terms “wide viewpoints” and “large motion” are used but lack clear quantitative measures. Providing specific metrics or thresholds for these terms, such as angular range for viewpoints or displacement magnitude for motion, would help readers better understand how your dataset outperform than others.
3.The preliminary section in 3.1 is extensive. If this content is not the authors’ own contribution, a concise overview would be more effective, allowing readers to focus on the novel aspects of the paper.
4. DUSt3R requires multi-view images for point cloud reconstruction, similar to COLMAP. What scenarios does DUSt3R handle that COLMAP cannot? Identifying and explaining these differences would clarify DUSt3R’s advantages and justify its use over COLMAP. The author could provide more results to show DUSt3R outperform COLMAP.
5.If my understanding is incorrect, please let me know. Most objects in the dataset are rigid objects, meaning they do not undergo mesh deformations like humans or animals do during motion, which makes the dataset less challenging. The authors could include some non-rigid objects in the dataset or discussing why you don't do that in the paper.

---

### Official Review · Reviewer_pnoQ · 2024-11-03

**Soundness:** 2
**Presentation:** 2
**Contribution:** 2
**Rating:** 5
**Confidence:** 3

**Summary:**

This paper introduces SC-4DGS, a pose-free optimization pipeline for dynamic view synthesis by combining DUSt3R for initial self-calibration, DynMF for dynamic motion representation, and a novel batchwise optimization strategy. The method claims to eliminate the need for structure-from-motion and enables the joint optimization of camera poses and dynamic scene representations, achieving high-quality rendering for complex dynamic scenes in a monocular setting. It also introduces a new dataset, Kubric-MRig, to evaluate both camera calibration and novel view synthesis performance in dynamic scenes.

**Strengths:**

The paper presents a fully integrated pipeline that allows for pose-free dynamic Gaussian splatting rendering.

Based on my understanding, this is more of a pipeline paper rather than a structure-based improvement. However, the batch-wise optimization introduces novelty by addressing the memory and computational limitations of DUSt3R, making the method scalable for dynamic scenes.

A new dataset that addresses limitations that previous benchmark could not assess

**Weaknesses:**

The proposed method heavily depends on prior approaches such as DUSt3R for both pose estimation and dynamic rendering. As a result, any failure in DUSt3R directly causes the failure of the proposed method. Moreover, the paper lacks some critical comparative experiments. While it repeatedly claims that structure-from-motion (SfM) methods, like Colmap, often fail, leading to inaccurate pose estimation and rendering, it would be beneficial to include experiments that clearly show where SfM combined with standard 4DGS falls short and where SC-4DGS excels. This would more effectively demonstrate the advantages of the proposed approach.

Additionally, the paper feels somewhat incomplete and lacks polish in its writing. For example, in the ablation study on pose initialization strategies, the following statement is problematic: "We explore four batch sampling strategies when initializing poses via DUSt3R: naive, sequential (SQ), strided batch (SB), and ours… SB uses the strided batch technique for optimization." This sentence is uninformative, as it merely reiterates the meaning of the abbreviation without adding value. Furthermore, the following sentence refers to "Appendix 4.4" for more details, but upon checking, there is no such appendix.

**Questions:**

Q1. The batchwise optimization makes joint optimization of pose and dynamic rendering possible, but is it capable of correcting pose estimation errors if DUSt3R provides a suboptimal initial pose estimation? In other words, how robust is the system to inaccuracies in the initial poses provided by DUSt3R?
Q2. If SC-4DGS shares similar failure conditions with most SfM-based methods, it seems like the approach is merely combining a two-step process into one without addressing the inherent issue of pose estimation errors. In scenarios where ground truth poses are not available, what specific conditions would lead to failures in pose estimation and rendering for SfM-based methods, but would still allow SC-4DGS to correctly render dynamic scenes and estimate poses?

---

### Official Review · Reviewer_rwSb · 2024-11-03

**Soundness:** 3
**Presentation:** 2
**Contribution:** 2
**Rating:** 3
**Confidence:** 5

**Summary:**

The paper presents SC-4DGS (Self-Calibrating 4D Gaussian Splatting), a novel approach for pose-free dynamic view synthesis from monocular videos. Current dynamic neural field models rely heavily on Structure from Motion (SfM) and static scene assumptions, which limit their applicability in real-world scenarios with significant motion. SC-4DGS addresses these limitations by jointly optimizing camera poses and dynamic Gaussian representations without requiring pre-calibrated camera data or static scenes.

The approach leverages DUSt3R, a geometric foundation model, to provide initial pose and point cloud estimations. It introduces batch-wise optimization and an extended motion representation tailored to DUSt3R’s capabilities, allowing the model to handle dense frames efficiently. SC-4DGS also incorporates several regularization terms to improve geometric accuracy in rendering.

To evaluate SC-4DGS, the authors introduce Kubric-MRig, a new benchmark dataset designed to test calibration and rendering performance in dynamic scenes with extensive object and camera motion. Experimental results show that SC-4DGS outperforms previous pose-free dynamic neural fields and achieves competitive results against state-of-the-art pose-free 3D neural fields.

**Strengths:**

The paper presents a significant advancement by proposing a batchwise optimization method for DUSt3R, enabling it to be used effectively for dynamic Gaussian Splatting (DGS) without relying on Structure from Motion (SfM) pipelines like COLMAP. This innovation is particularly valuable as it removes the dependency on pre-calibrated camera data, allowing the model to perform self-calibration even in scenes with extensive object and camera motion. By introducing this self-calibrating approach, the paper expands the application range of DGS, addressing a major gap in pose-free dynamic view synthesis.

**Weaknesses:**

Clarity of Figures and Tables:
The visual presentation of some figures and tables could be significantly improved for readability and clarity: Figure 1 is not illustrative and hard to follow even if combining the main content. At least it can add a legend or annotations that explain each variable clearly, making it challenging for readers to follow without detailed reference to the text. Adding a legend or in-figure labels could improve comprehension. Figure 4 would benefit from a horizontal arrangement of camera rotations, as it would make the sequence and rotation dynamics easier to interpret at a glance. Addressing these issues would make the paper more accessible, particularly in sections introducing new methods and experimental setups.

Excessive Detail in Preliminary and Method Sections (Sections 3.1 and 3.3):
Section 3.1 delves too deeply into the basics of 3D Gaussian Splatting, including eight detailed equations that do not play a critical role in subsequent sections. While introductory information is helpful, streamlining this part to focus only on the essentials would allow readers to concentrate on the novel contributions. Similarly, Section 3.3 introduces multiple variables in equations that are only referenced once, which adds cognitive load without enhancing understanding. Reducing or simplifying these equations, or moving some details to an appendix, could maintain the paper’s technical rigor while improving readability.

Dataset Design Limitations:
The Kubric-MRig dataset, while useful, is overly simplified in certain respects: The background lacks detail and depth variation, as it is primarily a simple ground plane. This simplification limits the model's ability to generalize to more complex real-world scenes where background intricacies affect depth and spatial perception. The dataset's objects are all synthetic geometric shapes or scanned objects with basic forms, which fall short of representing realistic, complex shapes such as human bodies, animals, or vehicles. Incorporating more diverse objects with varied textures and structures would enhance the dataset’s utility as a benchmark for dynamic view synthesis in practical applications. For example, including urban environments with buildings, natural scenes with vegetation, or indoor settings with furniture. For objects, adding articulated models of humans or animals, or complex mechanical objects like vehicles.

Limited Novelty:
While the batch-wise optimization approach for DUSt3R to enable its use in dynamic Gaussian Splatting is a valuable engineering contribution, the novelty is relatively incremental: The main contribution is an engineering improvement rather than a theoretical innovation, which may not meet the high standards of groundbreaking novelty expected at ICLR.
On the other hand, to strengthen the impact, the authors could discuss broader implications of this method, such as potential extensions or applications in other areas, or propose future directions that could build on this optimization. For example, it can be tested on other types of 3D reconstruction models including dynamic NeRFs which also rely on COLMAP, or if it has potential applications in fields like robotics or augmented reality.

**Questions:**

I'm wondering about the setup of the ground truth data and novel view data in the proposed dataset:
1. only 1-3 objects are dynamic in the synthesis space yet all the other 10-20 objects and backgrounds are still. Is it a reasonable ratio? if the still objects (including the simple background) are less challenging, will the error from the dynamic object be overwhelmed? Especially when the input camera poses go around the scene like a scanning, the still objects will get much benefit from this camera movement. It is recommended to justify this ratio of dynamic to static objects, perhaps based on real-world scenarios or existing datasets. Another idea is to conduct an ablation study varying this ratio to show its impact on the model's performance.

2. In the appendix, it is described that the monocular input ground truth is the 100 frames from the first camera, and the novel view ground truth is the 100 cameras at time 0. Is the novel view evaluation only tested on time 0? Won't introduce bias? Please clarify if it has been considered to evaluate novel views at different time steps. Or at least discuss potential biases this evaluation method might introduce and how they might address or mitigate these biases.

---

### Note · Authors · 2024-11-15

**Comment:**

Withdrawing the paper to enhance the contribution.

**Withdrawal Confirmation:**

I have read and agree with the venue's withdrawal policy on behalf of myself and my co-authors.